# Performance Analysis of Photovoltaic Integrated Shading Devices (PVSDs) and Semi-Transparent Photovoltaic (STPV) Devices Retrofitted to a Prototype Office Building in a Hot Desert Climate

**Abdelhakim Mesloub [1,\*]**, **Aritra Ghosh [2,3,4,\*]**, **Mabrouk Touahmia [1]**,
**Ghazy Abdullah Albaqawy [1]**, **Emad Noaime [1]** and **Badr M. Alsolami [5]**

[1] Department of Architectural Engineering, Ha'il University, Ha'il 2440, Saudi Arabia;
m.touahmia@uoh.edu.sa (M.T.); g.albaqawy@uoh.edu.sa (G.A.A.); e.noaime@uoh.edu.sa (E.N.)

[2] Environment and Sustainability Institute, University of Exeter, Penryn, Cornwall TR10 9FE, UK

[3] College of Engineering, Mathematics and Physical Sciences, Renewable Energy, University of Exeter,
Cornwall TR10 9FE, UK

[4] Renewable Energy, Stella Turk Building, University of Exeter, Penryn, Cornwall TR10 9FE, UK

[5] Islamic Architecture Department, College of Engineering and Islamic Architecture, Umm Al-Qura
University, Mekkah 21955, Saudi Arabia; bmsolami@uqu.edu.sa

\* Correspondence: a.maslub@uoh.edu.sa (A.M.); a.ghosh@exeter.ac.uk (A.G.)

**Abstract:** This paper presents the impact on energy performance and visual comfort of retrofitting photovoltaic integrated shading devices (PVSDs) to the façade of a prototype office building in a hot desert climate. EnergyPlus™ and the DIVA-for-Rhino© plug-ins were used to perform numerical simulations and parametric analyses examining the energy performance and visual comfort of five configurations, namely: (1) inclined single panel PVSDs, (2) unfilled eggcrate PVSDs, (3) a louvre PVSD of ten slats tilted 30° outward, (4) a louvre PVSD of five slats tilted 30° outward, and (5) an STPV module with 20% transparency which were then compared to a reference office building (ROB) model. The field measurements of an off-grid system at various tilt angles provided an optimum tilt angle of 30°. A 30° tilt was then integrated into some of the PVSD designs. The results revealed that the integration of PVSDs significantly improved overall energy performance and reduced glare. The unfilled eggcrate PVSD did not only have the highest conversion efficiency at $\eta$ 20% but generated extra energy as well; an essential feature in the hot desert climate of Saudi Arabia.

**Keywords:** photovoltaic shading device (PVSD); overall energy; tilt angle; visual comfort; energy saving; hot desert climate

## 1. Introduction

Rapid population growth coupled with leaps in urbanisation and industrialisation over the last two decades have exponentially increased electricity demand in the Kingdom of Saudi Arabia [1–3]. In the north-western province of Ha'il, domestic energy consumption has skyrocketed to unprecedented levels, leading to increasingly frequent power outages during the harsh summer months, and sometimes during the unusually cold winters as well, when power demands peak. At present, the Kingdom is in the midst of implementing its ambitious Saudi Vision 2030 programme; an undertaking that seeks to drive economic development by building a thriving, diversified, and sustainable economy [4,5]. However, satisfying increasing domestic and industrial electricity demands while keeping greenhouse gas emissions low is a major challenge. As the present Saudi administration recognises the need

to reduce the environmental impact of non-renewable energy sources, more consideration has been given to exploring alternative and more sustainable energy sources to improve the energy efficiency of buildings. Taking into account Saudi Arabia's abundance of solar radiance as well as its desert climate, photovoltaic (PV) technology could play a vital role in overcoming the energy issues of this hot and arid country [6]. As solar energy is easily converted into heat and electricity, it could power a significant proportion of domestic heating, ventilation, and air-conditioning (HVAC) systems, hot-water systems, lighting, and other key utilities.

The appeal of building-integrated photovoltaic (BIPV) systems is their ability to generate sustainable electricity simply by retrofitting building envelopes while simultaneously improving visual comfort [7–9]. Available in a variety of shapes, sizes, and designs, photovoltaic systems (PVs) blend seamlessly with commonly-used opaque and semi-transparent materials in architecture such as glass or metal, making them easy to integrate into every part of a building's envelope [9]. As such, PV systems such as photovoltaic integrated shading devices (PVSDs) are usually incorporated into building facades. A common trend in BIPV systems is the integration of different PVSD configurations into a facade, which serves as a new external skin for new and old buildings alike [10,11]. PVSD types include inclined single panel [12,13], outward tilted horizontal louvres [14,15], and unfilled eggcrate [16], as well as STPV modules [17] which are made of either crystalline silicon [18] thin-film PV [19], perovskite [20,21], or DSSC [22,23]. It is one of the more aesthetically pleasing architectural methods of converting excessive solar energy to electricity [24,25].

A properly-designed PVSD system is capable of effectively saving energy, improving thermal comfort, and reducing the glare of buildings [26,27]. Wienold et al. [28] found that internal fixed shading devices had lower performance than their external counterparts and were less economical than dynamic systems. As the angle of inclination is an important parameter in evaluating PVSD performance, Zhang et al. [29] utilised a numerical simulation model to study the benefits and energy saving capabilities of PVSDs fixed at numerous tilt angles and orientations in Hong Kong. They discovered that a southern orientation and a 20° tilt produced more overall energy benefits annually than interior blinds. An investigation of the thermal performance of PVSDs in winter by Yoo and Manz [30] found that they could be used as a double-envelope and insulation. Using DIVA Grasshopper 3D© and the window-to-wall ratio (WWR) variable, Settino et al. [28] developed an evolutionary algorithm to optimise the size and shape of shading systems in three European cities. It found that an optimally designed PVSD system could reduce annual energy consumption by up to 42%. Jayathissa et al. [31] examined the use of a dynamic integrated PV shading system to simultaneously optimise a building's energy demand and its PV energy output. Using a scale model and simulations to evaluate the performance and effect of louvre-integrated PV devices on indoor daylighting, Kim et al. [32] discovered that tilting the slats of a horizontal louvre to the downward position increased energy output and decreased indoor daylight levels. Sung et al. [33] developed louvre-integrated PV shading to enhance energy production and indoor visual comfort. Bellia et al. [34] investigated the influence of an external shading system on the energy consumption, space cooling, heating, and lighting of an ordinary Italian office. Khoroshiltseva et al. [35] used multi-objective optimisation analysis to design a versatile and efficient shading system that addressed increased energy demands in the winter for heating and lighting as well as cooling in the summer.

At present, because of the higher building energy consumption [36,37], powering buildings from PV is gaining importance. Imam et al. [38] investigated a grid-connected PV for building and from techno economic analysis it was found that 12.25 kW is the minimum requirement for a typical Saudi apartment. Lopez-Ruiz et al. [39] studied the potential of building roof top solar systems for Saudi Arabian architecture, which rarely employs the use of PVSDs. Only a few studies have investigated the overall energy performance of external fixed PVSDs over conventional shading devices [40] or examined the vertical and horizontal photovoltaic shading device in terms of insolation [4]. Therefore, this study performed a case study of the overall energy performance and visual comfort of five different configurations on a case study reference office building (ROB); the College of Engineering

building of the University of Ha'il, which is located in a hot desert. The EnergyPlus™ programme [41] and the DIVA-for-Rhino© software plug-in [11,42] were used to model the ROB's HVAC system, daylighting, and solar gains as well as the energy produced as a consequence of PVSD integration. The maximum energy output of solar PVs tilted at various angles and degrees was analysed through experimental investigation with an off-grid system. Finally, the energy saving potential of prototype small offices retrofitted with each of the five configurations was analysed and compared to that of the ROB, an average Saudi Arabian office building.

## 2. Methodology

Numerical parametric simulations were used as a data collection tool to perform quantitative and qualitative analyses. The following sections present the research materials and methods implemented in the study.

### 2.1. Case Study (ROB) and Climate

The prototype small office model (Figure 1c) was a curtain wall design with a fixed window-to-wall ratio (WWR) of 30%. The model featured two windows recessed at a depth of 30 cm and fitted with argon gas-filled double-glazed window panes coated with low emissive (low-E) glass that has a visible light transmittance (VLT) of 0.79. SketchUp™ was used to design the prototype small office model before it was exported to EnergyPlus™ to assess the energy saving potential of each configuration. This data was then compared to a case study reference office building (ROB), the College of Engineering building at the University of Ha'il. Built on flat ground with no shading from adjacent buildings, the ROB is 2.80 x 4.60 m and 3.0 m in height, lateral in typology, and has south-facing office units on the second floor that are separated by a central corridor.

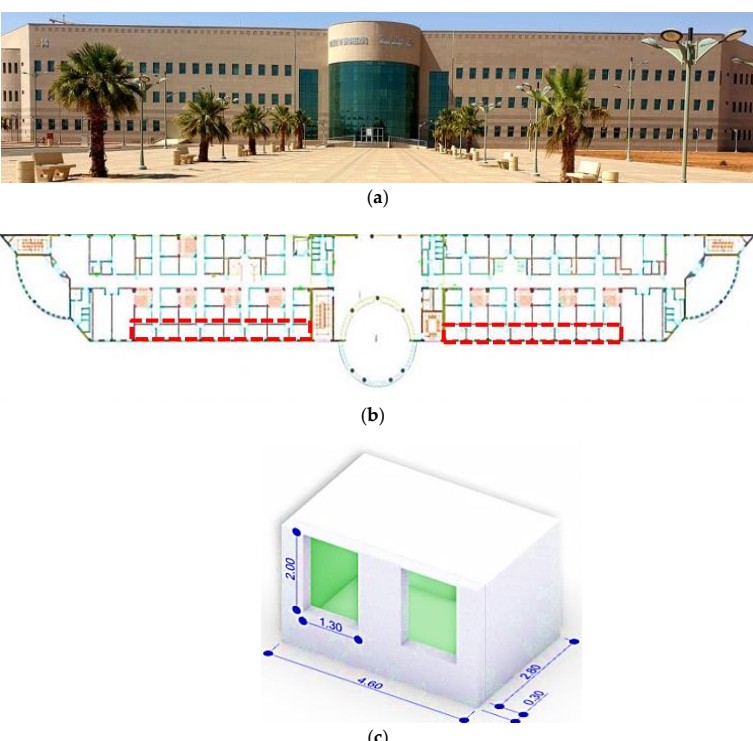

(a)

(b)

(c)

**Figure 1.** (**a**) The southern façade of the case study reference office building (ROB), the College of Engineering at the University of Ha'il (UoH); (**b**) the location of the tested offices in the ROB; and (**c**) a prototype model of a small office.

Situated at 27°31′ N latitude and 41°41′ E longitude, according to the Köppen climate classification, the location of our chosen ROB is a dry desert with a mean temperature of 31.1 °C in August (warmest month) and 10.6 °C in January (coolest month) while the average annual temperature is 20.9 °C (Figure 2). Its location also provides lots of solar radiance and daylight throughout the year for free. Therefore, it is preferable to utilise maximum daylighting in the building during the daylight hours. As such, a fully clear sky was selected as the standard design in the prototypes.

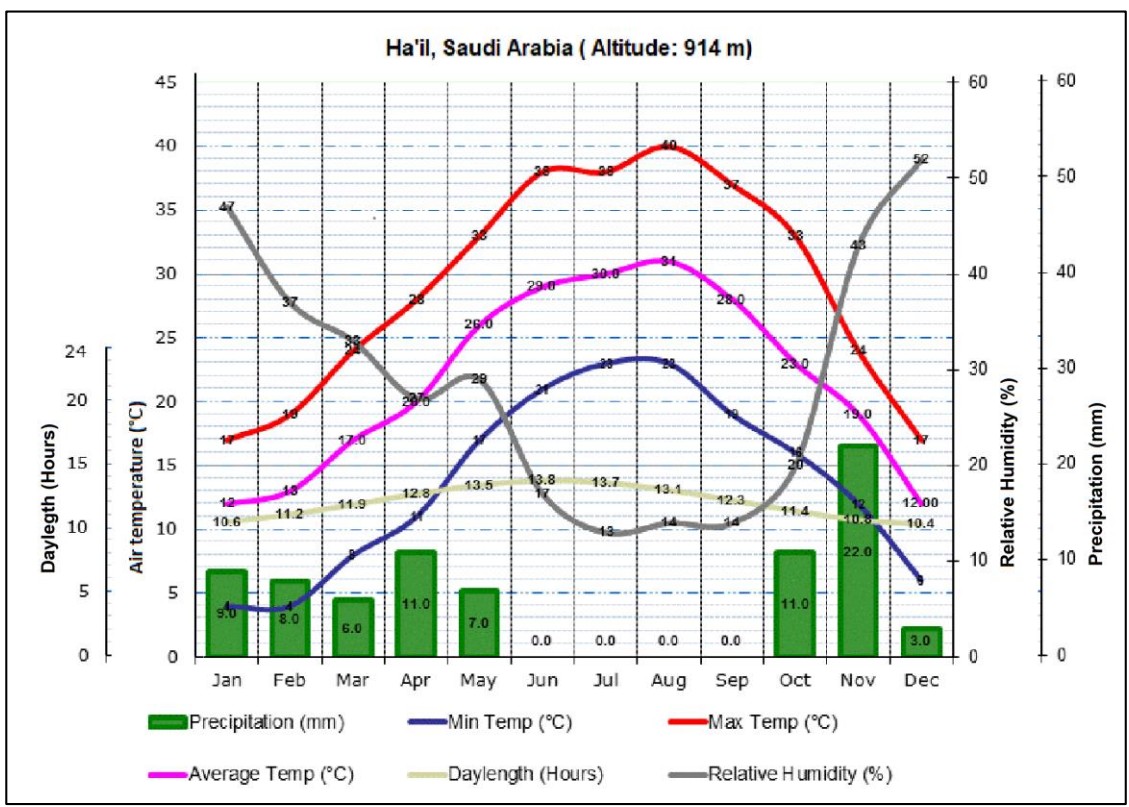

**Figure 2.** Monthly climatic data of Ha'il city.

## 2.2. PVSD Configurations

This study mainly focused on the overall energy performance of prototype small offices retrofitted with each of the five configurations (Table 1). As the angles, dimensions, and design of these configurations needed to be able to be retrofitted to building fenestrations to achieve maximum energy savings, uniformity of daylighting to reduce artificial lighting energy consumption, and maximum energy production for cooling and heating when required, the following five configurations were examined:

1.  An inclined single panel PVSD.
2.  An unfilled eggcrate PVSD.
3.  A louvre of ten outward-tilted slats.
4.  A louvre of five outward-tilted slats.
5.  A semi-transparent photovoltaic (STPV) module with 20% transparency.

**Table 1.** The different PVSDs configurations used in the simulations.

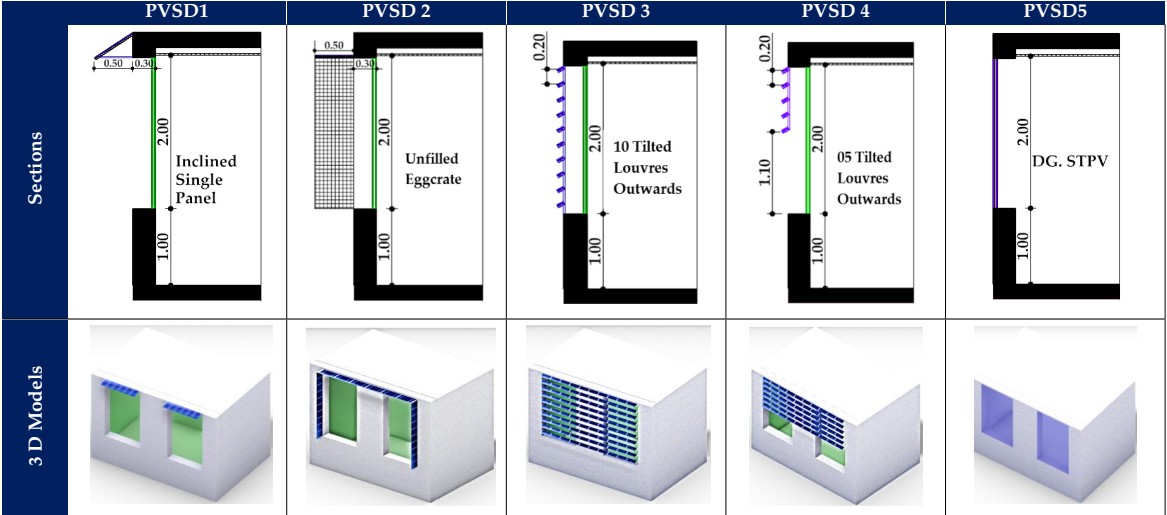

The curtain wall 3D model of each configuration featured two 30 cm-recessed windows. The inclined single panel configuration (PVSD1) was designed to overhang the tops of both windows by 50 cm without spacing and to tilt downward by 30° to avoid direct sunlight during excessively hot periods. In the unfilled eggcrate configuration (PVSD2), one horizontal 50 cm-photovoltaic panel skirted the top of both windows while two vertical 50 cm-photovoltaic panels bordered the left and right sides. In PVSD3, a louvre of ten 10 cm-slats was tilted downward at a 30° angle and spaced 20 cm apart to avoid mutual shading throughout the day. This layout was duplicated in PVSD4 with only five slats across the top half of both windows to improve the users' field of vision. Finally, PVSD5 placed a semi-transparent photovoltaic (STPV) module, with 20% transparency, at the outer edge of both windows to prevent shaded areas in the PV panel. These PVSDs were not connected to grid thus no grid inverters were employed for study. Additionally, because of the nature of the PVSD, no inverters were considered in this study. Prime focuses were to find the building energy saving through exploiting daylighting, controlling solar heat gain, and generating DC power.

*2.3. PV Energy Output Simulations*

The meteorological data of the city of Ha'il was extracted from the Meteonorm© database and used in the EnergyPlus™ simulations. EnergyPlus™ "Equivalent One-Diode model"; which uses an empirical relationship to predict the operating performance of the PV based on conditions such as the temperature of the PV cell, and estimate the conversion efficiency of each time-step empirically (that cannot be determined directly through physical measurements), was then used to estimate the irradiation of tilted surfaces. The simulation programme used data from the manufacturer's catalogue (Table 2) to calculate these values.

**Table 2.** Electrical characteristics under standard test conditions (STC) of different PV types.

| Electrical Properties (STC) | | |
|---|---|---|
| Array Types | PVSD Multi-Crystalline | STPV Thin-Film (Amorphous) |
| Dimension (length, width, thickness) | 767 x 502 mm | 2000 x 1300 mm |
| Max power (Pmax) | 40 watt | 88.4 watt |
| Efficiency of module ($\eta$) | 10% | 3.4% |
| Max power voltage (Vpm) | 21 V | 78 V |
| Max power current (Ipm) | 2.37 A | 1.15 A |
| Open circuit voltage | 21 V | 100 V |
| Short circuit current | 2.58 A | 1.43 A |

### 2.4. Experiment Validation of PV Energy Output

At present, energy model simulation tools are used with greater frequency to measure energy output. Although most of these simulation tools employ standard sky types, as defined by the Commission Internationale de l'Éclairage (CIE), their accuracy and applicability in the Saudi Arabian climate warrants validation.

As such, the energy output of an off-grid system was measured. The system was set up according to the VDAS® (Versatile Data Acquisition System); a standard arrangement of a multi-crystalline photovoltaic panel array, battery storage, and a solar charge controller (Figure 3). Measurements were taken under clear skies in spring, between 15 and 18 March 2020. These measured values were then compared to the values derived using EnergyPlus™ to validate PV energy outputs. The American Society of Heating, Refrigerating and Air-Conditioning Engineers (ASHRAE)eveloped Guideline 14 strongly recommends that the mean bias error (MBE) should be less than 10% while the coefficient of variation of the root mean square error (CV(RMSE)) should be less than 30% in order to validate a PVSD energy model [43].

$$\mathbf{MBE}(\%) \; = \; \frac{\sum_{i=1}^{Np}(mi - si)}{\sum_{i=1}^{Np}(mi)} \tag{1}$$

$$\mathbf{CV\ RMSE} \; = \; \frac{\sqrt{\sum_{i=1}^{Np}(mi - si)^2/Np}}{m} \tag{2}$$

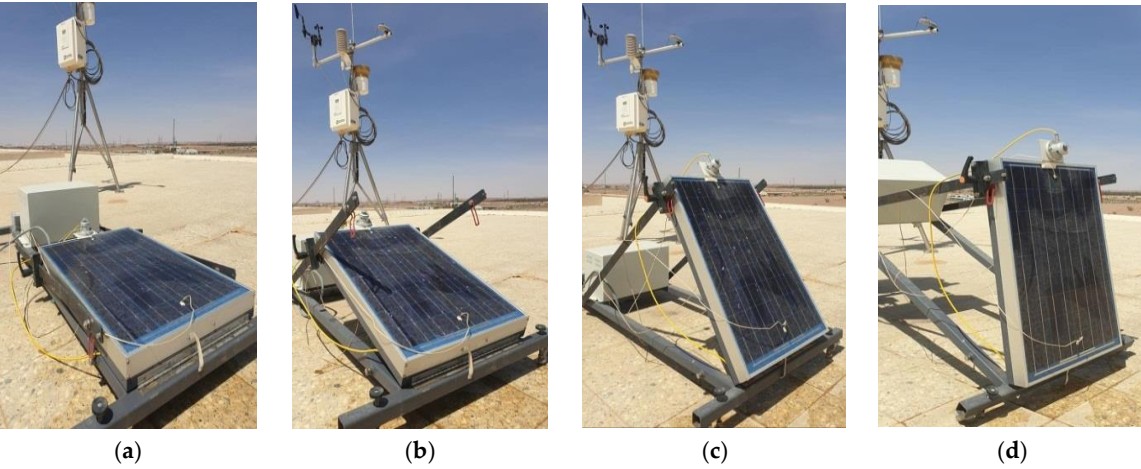

| (a) | (b) | (c) | (d) |

**Figure 3.** Field measurements of an off-grid PV system with different tilt angles: (**a**) horizontal angle; (**b**) tilted with 30°; (**c**) tilted with 60°; (**d**) tilted with 90°.

### 2.5. Thermal Simulation

Dynamic thermal simulations were performed using the Ideal Loads Air Systems (ILAS); an ideal HVAC system with a coefficient of performance (COP) of one. To satisfy the accepted operative temperature of a new office building as well as the required zone temperature set-point as per ISO 7730, the ILAS HVAC was assumed to supply sufficient cooling and heating at 20 °C and 26 °C air temperature, respectively [44]. As the regulation policy of Saudi Arabia stipulates that heating and cooling systems are only to be turned on during working hours, the occupancy schedule of the office building was set to between 08:00 and 17:00 from Sunday to Thursday and considered closed on weekends and bank holidays. The construction materials of the office building, as well as their thermo-physical properties, were exported to OpenStudio® for building energy analysis. Table 3 shows extracts of optical properties as taken from WINDOW 7.7, a windows and daylighting software.

**Table 3.** Thermal properties of external wall building and glazing layers.

| Material (External Wall Layers) | k (W/m.K) | ρ (kg/m³) | c (J/kg.K) |
|---|---|---|---|
| H.W.H.C.B. (150 mm) | 0.96 | 1362 | 879 |
| Thermal insulation (30 mm) | 0.04 | 91 | 837 |
| H.W.H.C.B. (100 mm) | 0.81 | 1618 | 879 |
| Gypsum board (19 mm) | 0.16 | 785 | 1090 |
| **Glazing (Optical and Thermal properties)** | **VLT** | **U-value** | **SHGC** |
| Double glazing low-E - Argon gas | 0.79 | 1.10 | 0.65 |
| Double glazing semi-transparent PV | 0.20 | 2.70 | 0.14 |

## 2.6. Daylighting and Lighting Energy Simulations

In the artificial lighting simulations, a photosensor was used to dim the activated lighting until the work plane illuminance reached the desired value of 300 lx. However, artificial lighting was assumed to be switched on during the occupancy hours of 08:00 to 17:00. Their operating profiles follow their real use. To evaluate indoor visual comfort, the DIVA-for-Rhino© plug-in was used to obtain daylight autonomy (DA) and useful daylight illuminance (UDI) metrics [45]. The UDI bin was set to 100 to 2000 lx while the DA threshold was set to 300 lx. Glare analyses were also performed by calculating the daylight glare probability (DGP) and the 3D illuminance contour map of each configuration. The international standard (Table 4) was selected as the performance indicator.

**Table 4.** The performance indicators of visual comfort used in this study.

| Analysis | Criteria | Performance Indicator |
|---|---|---|
| | UDI | 100 lux < Dark area (need artificial light) |
| | | 100 lux–2000 lux (comfortable), at least 50% of the time |
| | | >2000 lux too bright with thermal discomfort |
| **Quantitative + qualitative** | DA | Set up 300 lx |
| | WPI | WPI recommended 300–500 lux |
| | | 0.35 < imperceptible glare |
| | DGP | 0.35–0.40 perceptible glare |
| | | 0.4–0.45 disturbing glare |
| | | >0.45 intolerable glare |

## 3. Results and Discussion

### 3.1. Analysis and Validation of Energy Outputs at Different Tilt Angles

For greater accuracy, the energy output of the off-grid system and EnergyPlus™ data at different tilt angles—0° (horizontal), 30°, 60°, and 90° (vertical)—were measured hourly over four different days and compared (Figure 4). The energy output of the EnergyPlus™ simulations was slightly higher for all tilt angles than the off-grid system results, particularly at midday. A maximum energy output of 38 W/h was observed at 1.00 p.m. when the panel was titled at 30°. This proved the validity and reliability of the EnergyPlus™ simulations as the MBE of the 0°, 30°, 60°, and 90° tilted models were between 5.4%, 6%, 9.2%, and 11%, respectively, while the CV(RMSE) of the 0° tilted model was 7.28% and 12.45% for the 90° tilted model.

Figure 5 presents the hourly energy output of the angles over four different seasons. As the PV energy output at 0° tilt was generally higher than the 90° tilt, therefore, it could be used as a PVSD. The 30°-tilt PV produced the highest amount of electricity daily in all seasons except summer while the 0°-tilt PV produce more electricity overall as it had a southern orientation and due to the sun path position, therefore, received more solar radiance. It is remarkable that the value of indoor illuminance (lux) reduced in the evening period of 21 of June because of reduction of external illuminance, where the sky condition changed from clear sky to an overcast sky. The annual energy output of the 30°-tilt PV was up to 91.7 kWh/year (Appendix A). It is also noteworthy that the daily energy output of the 90°-tilt PV was only 12% to 43% compared to the 30°-tilt PV in all seasons. As such, a 30° tilt was integrated into the design of the inclined single panel, tilted 10-slat louvre, and tilted 5-slat louvre PVSDs.

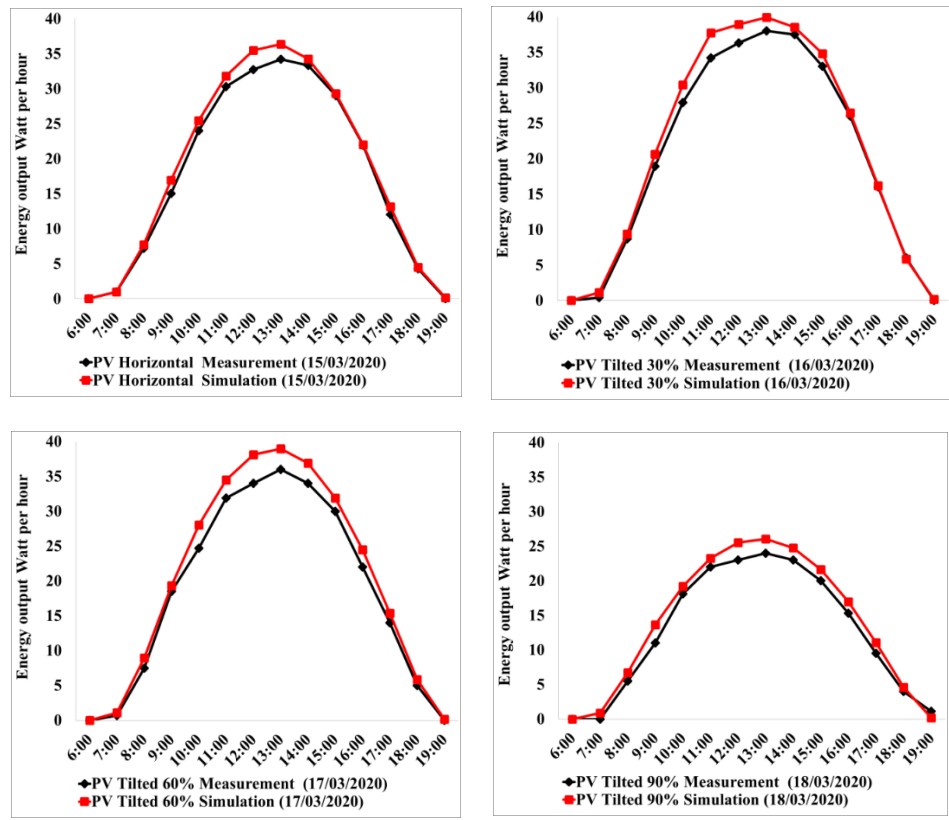

**Figure 4.** Validation of the energy output of the 0°, 30°, 60°, and 90° tilted models over four days in spring.

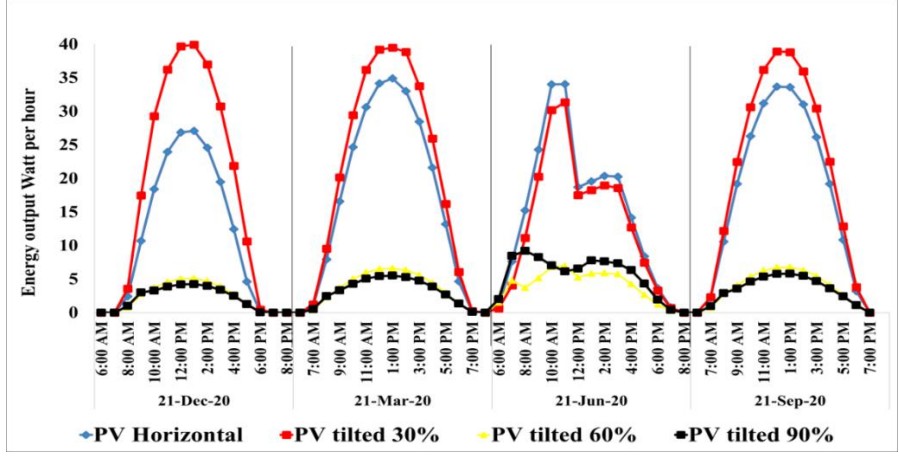

**Figure 5.** The hourly energy output of the different tilt angles across four seasons.

### 3.2. Energy Performance

EnergyPlus™ used Ha'il city's hourly weather data from the Meteonorm© database to simulate the annual energy performance of the different PVSDs. Figure 6 presents the cooling and heating loads, artificial lighting, energy consumption, as well as the energy generated by the ROB, PVSDs, and the STPV module.

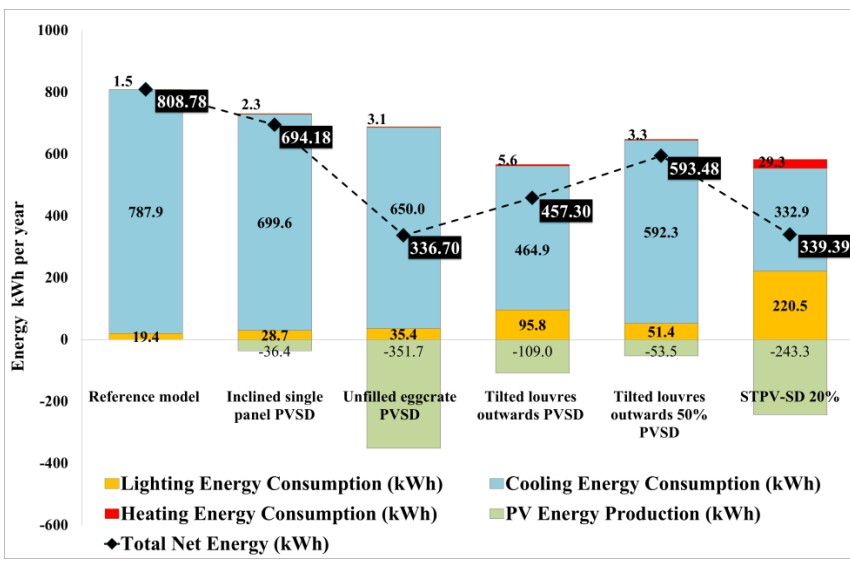

**Figure 6.** Yearly overall energy performance of the different configurations.

The simulation results indicated that the ROB and the configurations expended the most energy on cooling annually. The ROB utilised 97.5% of its total net energy on cooling while the inclined single panel, unfilled eggcrate, tilted 10-slat louvre, and tilted 5-slat louvre PVSDs consumed only 2%, 4%, 21%, and 9.24% less energy on cooling, respectively. However, the solar heat gain coefficient (SHGC) and heat energy transferred through a window (U-value) (Table 2) of the double-low-E glass used in the STPV modules reduced its cooling energy consumption by up to 75%. In contrast, the heating energy consumption per annum was negligible and, at the most, did not exceed 6% in the STPV module due to a lack of solar transmission.

PVSDs were found to increase the annual lighting energy consumption by up to 21% compared to the ROB while the STPV module recorded the highest energy consumption (220.5 kWh) due to its low VLT rate, only 20%. This finding was corroborated by one other study [46].

The unfilled eggcrate PVSD had the highest amount of annual PV energy production (351.7 kWh) followed by the STPV module (243.3 kWh) as they received more solar radiance than the other PVSDs in winter. The energy output of both these configurations was able to satisfy approximately 50% of their respective energy demands. As the single inclined panel, tilted 10-slat, and 5-slat louvre PVSDs produced the least amount of energy, therefore, the addition of a 30° tilt to the panels did not generate more energy as it reduced the surface area exposed to solar radiance. The efficiency of the unfilled eggcrate, single inclined panel, tilted 10-slat, and 5-slat louvre PVSDs was approximately 10% each and 3.4% for the STPV module (Table 2).

The overall energy consumption (OEC) trend of the ROB indicated that the use of recessed windows in hot desert climates was counterproductive as it enabled more heat transfer (U) into the ROB. In contrast, the addition of these configurations (PVSDs) not only significantly reduced cooling energy consumption but produced a considerable amount of electricity as well. Although they increased heating and lighting energy consumption, this was negligible compared to their reduction in cooling loads. Therefore, a balanced and carefully selected optimum configuration should be applied to all buildings.

### 3.3. Daylighting and Visual Comfort

Figure 7 presents the mean daylight autonomy (DA) and useful daylight illuminance (UDI) metrics within the ROB and the prototype small office models of the configurations. The ROB had the highest mean DA300 lx and UDI 100 to 2000 lx percentages with only moderate visual discomfort. The 20% transparency of the STPV module did not meet the minimum requirements of the mean DA and

recorded 0% while its UDI < 100 lx exceeded 90% thereby requiring the use of artificial light throughout the occupancy hours. Only the inclined single panel PVSD exceeded the 50% threshold of the DA300 lx while all the configurations achieved the mean UDI of 100 to 2000 lx. Only the tilted 10-slat louvre PVSD and the STPV module were able to totally eliminate visual discomfort (UDI < 2000).

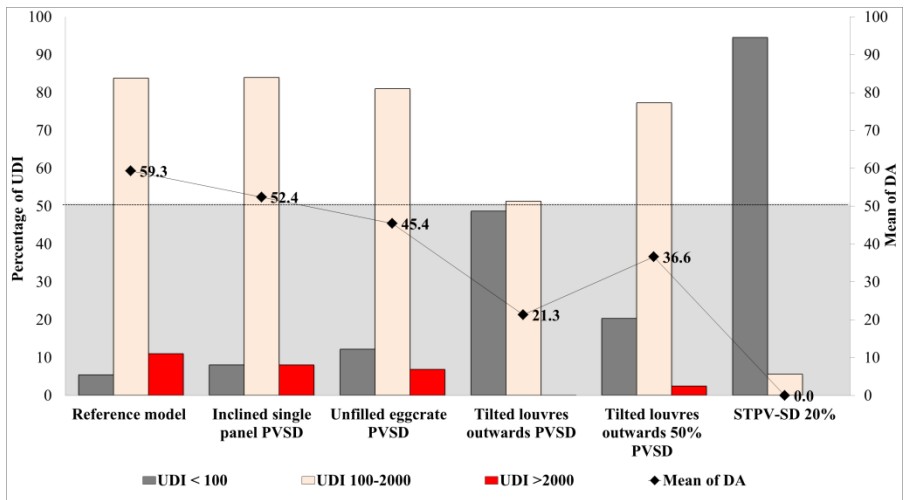

**Figure 7.** Climate-based analysis of annual mean daylight autonomy (DA) and useful daylight illuminance (UDI) of the ROB and the configurations.

### 3.4. Daylight Glare Probability

Table 5 shows the glare discomfort caused by the ROB and the configurations. Daylight glare probability (DGP), 3D illuminance contour map metrics, and mean illuminance were used to assess the glare of each configuration. As seen in Figure 8, the ROB had the lowest mean DGP; the perceptible and disturbing glare mainly appeared in winter season between 10 a.m. and 3 p.m. and the DGP value was between 27 and 40 over the four assessment days (refer to Appendix B for more details). The mean DGPs were within acceptable limits during the summer solstice and spring equinox but, during the winter solstice, disturbing glare was detected at 12:00 and perceptible glare at 09:00 and 15:00. The integration of a 30° tilt to the 10-slat and 5-slat louvre PVSDs decreased the contour shape during the winter solstice and produced an imperceptible glare with a mean illuminance ranging between 367 and 516 lx. The inclined panel PVSD had the least glare with a mean illuminance of between 765 and 2123 lx, which is higher than the recommended value. Overall, the configurations considerably enhanced visual comfort with a minimum 2° reduction in mean DGP compared to the ROB. Hence we can claim that PVSDs are suitable for winter season while STPVs with 20% of transparency totally eliminate the glare issue but do not provide sufficient indoor illuminance.

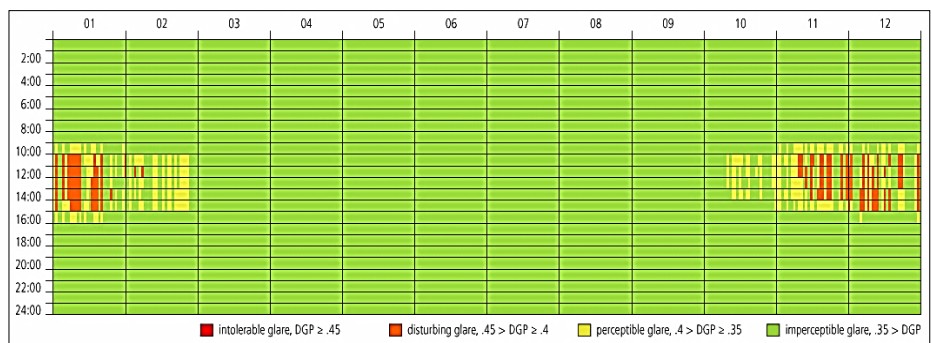

**Figure 8.** Annual daylight glare probability (DGP) analysis of ROB.

**Table 5.** The illuminance contour map of PVSDs configurations under clear sky at summer and winter solstice and spring equinox at (9.00 a.m., 12.00 p.m., 3.00 pm).

| | | ROB | PVSD1 | PVSD2 | PVSD3 | PVSD4 | STPV |
|---|---|---|---|---|---|---|---|
| **Solstice Winter (21 December)** | 09.00 A.M | | | | | | |
| | | DGP = 31 Mean ILL = 799 lx | DGP = 29 Mean ILL = 765 lx | DGP = 29 Mean ILL = 599 lx | DGP = 27 Mean ILL = 367 lx | DGP = 28 Mean ILL = 581 lx | DGP = 02 Mean ILL = 31 lx |
| | 12.00 P.M | | | | | | |
| | | DGP = 40 Mean ILL = 2582 lx | DGP = 38 Mean ILL = 2123 lx | DGP = 36 Mean ILL = 2103 lx | DGP = 30 Mean ILL = 516 lx | DGP = 32 Mean ILL = 1260 lx | DGP = 08 Mean ILL = 111 lx |
| | 3.00 P.M | | | | | | |
| | | DGP = 37 Mean ILL = 1082 lx | DGP = 35 Mean ILL = 1044 lx | DGP = 31 Mean ILL = 844 lx | DGP = 28 Mean ILL = 482 lx | DGP = 30 Mean ILL = 774 lx | DGP = 04 Mean ILL = 42 lx |

**Table 5.** *Cont.*

| | | ROB | PVSD1 | PVSD2 | PVSD3 | PVSD4 | STPV |
|---|---|---|---|---|---|---|---|
| | 09.00 A.M |  |  |  |  |  |  |
| | | DGP = 28 Mean ILL = 119 lx | DGP = 27 Mean ILL = 113 lx | DGP = 27 Mean ILL = 101 lx | DGP = 25 Mean ILL = 67 lx | DGP = 26 Mean ILL = 90 lx | DGP = 01 Mean ILL = 5 lx |
| Solstice Summer (21 June) | 12.00 P.M |  |  |  |  |  |  |
| | | DGP = 33 Mean ILL = 275 lx | DGP = 32 Mean ILL = 242 lx | DGP = 31 Mean ILL = 221 lx | DGP = 29 Mean ILL = 129 lx | DGP = 30 Mean ILL = 183 lx | DGP = 03 Mean ILL = 11 lx |
| | 3.00 P.M |  |  |  |  |  |  |
| | | DGP = 27 Mean ILL = 127 lx | DGP = 26 Mean ILL = 120 lx | DGP = 26 Mean ILL = 108 lx | DGP = 25 Mean ILL = 73 lx | DGP = 26 Mean ILL = 96 lx | DGP = 01 Mean ILL = 5 lx |

**Table 5.** *Cont.*

| | | ROB | PVSD1 | PVSD2 | PVSD3 | PVSD4 | STPV |
|---|---|---|---|---|---|---|---|
| **Equinox Spring (21 March)** | **09.00 A.M** |  |  |  |  |  |  |
| | | DGP = 28 Mean ILL = 263 lx | DGP = 27 Mean ILL = 228 lx | DGP = 27 Mean ILL = 155 lx | DGP = 25 Mean ILL = 82 lx | DGP = 26 Mean ILL = 140 lx | DGP = 01 Mean ILL = 8 lx |
| | **12.00 P.M** |  |  |  |  |  |  |
| | | DGP = 31 Mean ILL = 1133 lx | DGP = 30 Mean ILL = 721 lx | DGP = 30 Mean ILL = 688 lx | DGP = 27 Mean ILL = 103 lx | DGP = 28 Mean ILL = 250 lx | DGP = 02 Mean ILL = 35 lx |
| | **3.00 P.M** |  |  |  |  |  |  |
| | | DGP = 29 Mean ILL = 449 lx | DGP = 28 Mean ILL = 370 lx | DGP = 28 Mean ILL = 252 lx | DGP = 26 Mean ILL = 96 lx | DGP = 27 Mean ILL = 176 lx | DGP = 01 Mean ILL = 14 lx |

*3.5. Energy Saving Potential*

Figure 9 presents the energy saving potential of the PVSDs at varying efficiencies ($\eta$10%, $\eta$15% and $\eta$20%) and the STPV module at $\eta$3.1%. Although all the configurations had energy savings ranging between approximately 14.2% to plus-energy, the results were positive across the board in comparison to the ROB. The unfilled eggcrate PVSD saved the most energy, 58.4% to 101.9%, at conversion efficiencies of between $\eta$10% to $\eta$20%, respectively, as the horizontal and vertical configuration of its PV panels enabled it to receive maximum solar radiance, thereby producing more energy in addition to reducing its cooling energy consumption. Although the STPV module saved energy by 58%, it was at a very low conversion efficiency of $\eta$3.4%. This proved that thermal performance was more important than conversion efficiency. This configuration also did not meet minimum visual comfort requirements due to the low transparency of STPV window technology. It was discovered that the tilted 10-slat louvre PVSD could provide nearly twice the amount of energy savings than the tilted 5-slat louvre PVSD. However, the latter configuration omitted to install slats in the lower section of the window to provide users with a view of the outdoors. Finally, the inclined single panel PVSD had the worst energy saving potential of the five configurations. Therefore, a balanced trade-off between the thermal properties of glazing, the conversion efficiency of the PV panels, and visual comfort is vital to achieve a plus-energy building.

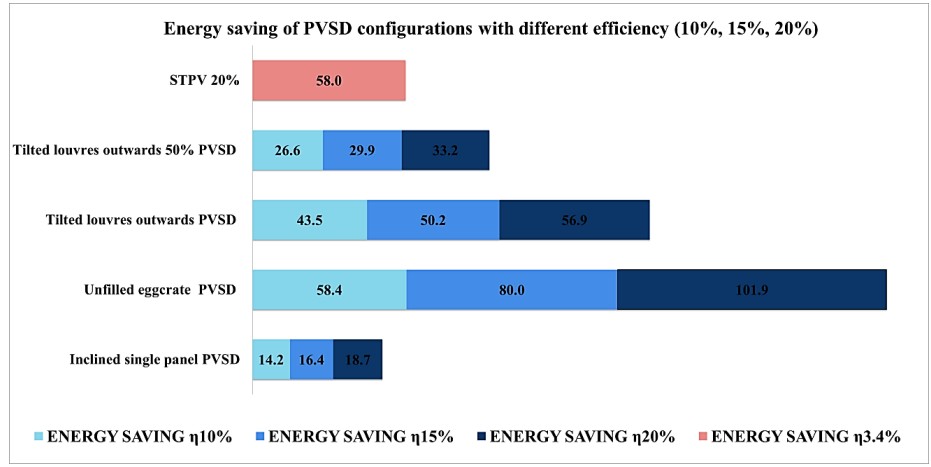

**Figure 9.** Energy saving percentages at different PV conversion efficiencies in comparison to the ROB.

## 4. Conclusions

This study examined for the first time the impact on overall energy performance and visual comfort of retrofitting four different PVSD configurations and one STPV module to a south-facing prototype small office model in the College of Engineering building of the University of Ha'il, under a hot desert climate, using Energy-plus and DIVA-for-Rhino© plug in building energy simulation tools. A comprehensive analysis of the yearly and hourly energy output of various tilt angles was first performed using an off-grid system to determine the optimum tilt angle, 30°, as well as to validate the data of our simulation models. A 30° tilt was integrated into the design of the inclined single panel, tilted ten-slat louvre, and tilted five-slat louvre PVSDs to attain maximum energy output. In order to select the optimum configuration, it is necessary to find a delicate balance between visual comfort and energy performance. The results of our simulations highlighted the following important points:

- Climate profoundly affected cooling and heating energy consumption as the integration of outward tilted slats in the louvre PVSDs as well as the double-low-E window pane of the STPV module significantly reduced cooling and heating energy consumption due to the thermal properties of the STPV window pane compared to double-low-E window panes applied in other configurations. Furthermore, it blocked direct solar radiance throughout the year.

- Although an effective configuration and daylight control strategy could eliminate glare discomfort, they increased lighting energy consumption, particularly in the STPV module.
- The unfilled eggcrate PVSD was the optimum configuration as it could produce plus energy by means of conversion efficiency ɳ= 20% while simultaneously providing visual comfort.

This study contributes to knowledge on PVSD and STPV module integration into office building envelopes in hot desert climates. Policymakers and architects may utilise the results of this research to retrofit PVSDs and STPV windows to conventional buildings in similar climates. However, future studies may focus on the return on investment (ROI) as well as other building typologies. In conclusion, the integration of PVSD systems and STPV modules into the building envelope offers large energy saving potential that could surpass zero net energy consumption, enhances visual comfort, and contributes to lower $CO_2$ emissions.

**Author Contributions:** Conceptualization, A.M., M.T. and A.G.; methodology, A.M.; software, A.M.; validation, A.M.; formal analysis, A.M.; investigation, A.M.; resources, A.M.; data curation, A.M. and A.G.; writing—original draft preparation, A.M.; writing—review and editing, A.G., M.T., G.A.A., B.M.A.; visualization, A.G., E.N. and M.T. All authors have read and agreed to the published version of the manuscript.

**Funding:** This research has been funded from the Research Deanship at University of Ha'il – Saudi Arabia through project number RG- 20 105.

**Conflicts of Interest:** The authors declare no conflict of interest.

## Appendix A

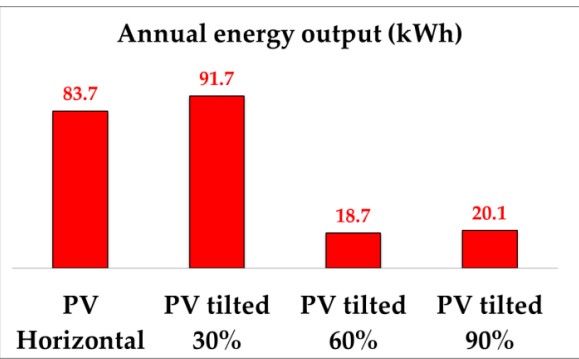

**Figure A1.** Annual energy output of different tilted angle of an off-grid PV system.

## Appendix B

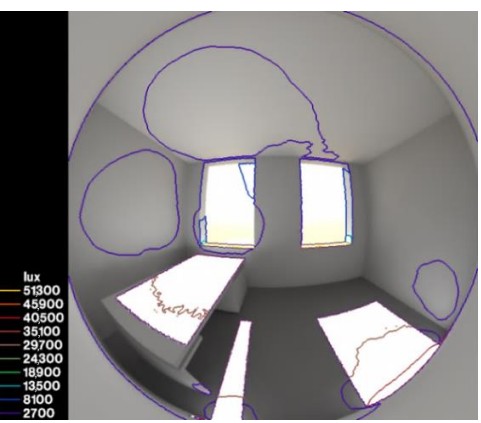 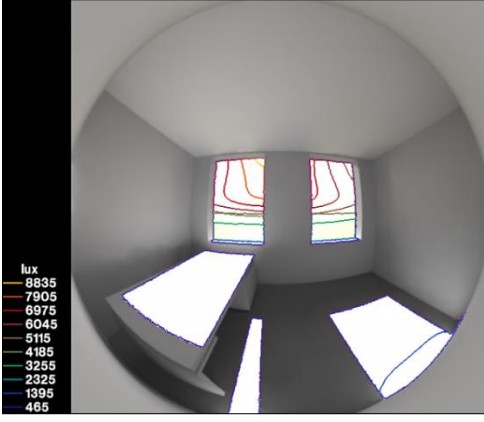

**Figure A2.** (The worst case) 3D illuminance contour map of the ROB compared with STPV configuration on winter solstice at midday.

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
