# Peer review of "Performance Analysis of Photovoltaic Integrated Shading Devices (PVSDs) and Semi-Transparent Photovoltaic (STPV) Devices Retrofitted to a Prototype Office Building in a Hot Desert Climate"

_sustainability, doi:10.3390/su122310145_

Round 1

Reviewer 1 Report

I want to congratulate the authors for the extremely interesting study.

Author Response

Thank you for Positive evaluation

Reviewer 2 Report

This paper is well-written and well-organized. Also, the topic is timely and since the manuscript contains experimental results, it seems very practical and interesting to readers.

Author Response

Thank you very much for your positive comments

Reviewer 3 Report

In introduction section, literature review can be extended and focused on other researchers methods of analysis and results in order to stress the novelty of your work in last paragraph (line 83).

 In section 2.1, where the case study is presented, it can be presented more information about PVSD. Particularly, PVSDs are connected to the grid or in an off-grid mode, there is an inverter?  A short description of the system.

 Simulations with energy plus take into account the PV system as grid-connected or off grid? In figure 6 PV energy production is provided to the grid or is used for lighting, heating and cooling demands. It is significant to clear up the inputs on energy plus.

In line 166, it can be defined which is the empirical relationship.

In line 311-312 PVSD efficiencies concern STC efficiency of PV Modules or something else?

In section 4 except from contribution it can be stressed more the novelty of your approach.

Author Response

CORRECTION OF REVIEWER COMMENTS

PAPER TITLE:  Performance Analysis of Photovoltaic Integrated Shading Devices (PVSDs) and Semi Transparent Photovoltaic (STPV) Retrofitted to a Prototype Office Building in a Hot Desert Climate

REVIEWER (2) COMMENTS

ACTION TAKEN

Reviewer (2) Comments

1

In introduction section, literature review can be extended and focused on other researchers methods of analysis and results in order to stress the novelty of your work in last paragraph (line 83).

The novelty has been highlited in the last paragraph:

At present, because of the higher building energy consumption [35, 36] powering building from PV is gaining importance. Imam, A et al. [37] investigated a grid connected PV for building and from techno economic analysis it was found that 12.25 kW is the minimum requirement for a typical Saudi apartment. Lopez-Ruiz, H. G et al. [38] studied the potential of building solar roof top for Saudi Arabian architecture rarely employs the use of PVSDs. Only a few studies have investigated the overall energy performance of external fixed PVSD over conventional shading devices [39]. Also, examined the vertical and horizontal photovoltaic shading device in term of insolation [4]. Therefore, this study performed a case study of the overall energy performance and visual comfort of five different configurations on a case study reference office building (ROB); the College of Engineering building of the University of Ha’il which is located in a hot desert.

2

In section 2.1, where the case study is presented, it can be presented more information about PVSD. Particularly, PVSDs are connected to the grid or in an off-grid mode, there is an inverter?  A short description of the system.

 These PVSDs are not connected to grid and no grid inverters are employed. Also, because of the nature of the PVSD, no inverters are considered in this study. Prime focuses were to find the building energy saving through exploiting daylighting, controlling solar heat gain and generates DC power.  

3

Simulations with energy plus take into account the PV system as grid-connected or off grid? In figure 6 PV energy production is provided to the grid or is used for lighting, heating and cooling demands. It is significant to clear up the inputs on energy plus.

- In the simulation part the system considered as an off grid system.

-The energy production from PV module with different configurations used to reduce the overall energy consumption (cooling, heating and lighting energy consumption).

4

In line 166, it can be defined which is the empirical relationship.

The EnergyPlus PV module employs equations for an empirical equivalent circuit model to predict the current-voltage characteristics of a single module.

5

In line 311-312 PVSD efficiencies concern STC efficiency of PV Modules or something else?

Yes, the efficiency of PV module used in the simulation is (n=10%) under STC condition. However, The efficiency with n=15% and 20% is available in market and proposed to see the possibility percentage of energy saving.

6

In section 4 except from contribution it can be stressed more the novelty of your approach.

Thank you and we now improved the conclusion:

This study examined for the first time the impact on overall energy performance and visual comfort of retrofitting four different PVSD configurations and one STPV module to a south-facing prototype small office model in the College of Engineering building of the University of Ha’il, under hot desert climate using Energy-plus and DIVA-for-Rhino© plug building energy simulation tool. A comprehensive analysis of the yearly and hourly energy output of various tilt angles was first performed using an off-grid system to determine the optimum tilt angle; 30⁰; as well as to validate the data of our simulation models. A 30⁰ tilt was integrated in the design of the inclined single panel, tilted ten-slat louvre, and tilted five-slat louvre PVSDs to attain maximum energy output. In order to select the most optimum configuration, it is necessary to find a delicate balance between visual comfort and energy performance. The results of our simulations highlighted the following important points:

Reviewer 4 Report

The authors present an analysis of different configurations of PV systems integrated into office buildings. For this analysis they take into account energy production, visual comfort, etc. The methodology is interesting and the results can contribute to the progress of science in this field. However, in my view, the authors should correct some aspects of the article before publication:

  • Review the list of authors: I think there is a lack of "," between the first two authors and it seems that there are surnames in lower case
  • Line 218: is the reference to table 4 correct? Authors should check the numbering of all tables and figures and their references in the text
  • Line 239: the authors could explain the higher levels of energy production on horizontal surfaces in summer by referring to the position of the sun
  • Line 292: a reference to a table 5 that does not exist in the document is made. Does it refer to table 4?
  • Table 4: the pictures included in this table are too small and cannot be clearly understood. Also, the explanatory texts in the first columns of the table cannot be seen properly.
  • Figure 1: For a better understanding of the working scenario, figure 1 should appear before table 1
  • Figure 2:
    • the source of the data must be indicated;
    • the vertical axis lists magnitudes that are not represented in the graph
  • Figure 4:
    • the authors say that they represent energy but the unit indicated on the vertical axis (watts) is not a unit of energy (but of power). The authors should review the energy units also in the paragraph before this figure.
    • The comparison of the influence of tilt angles would be clearer if it referred to the same day
  • Figure 5: Authors should specify whether the data represented are simulated or real. The authors could explain the shape of the graph on 21 June
  • Figure 8: the results shown in this figure and the conclusions drawn from them need to be further explained
  • Figure 9: The efficiency percentages shown in the figure do not correspond to those quoted in the text (lines 311-312)

Author Response

CORRECTION OF REVIEWER COMMENTS

PAPER TITLE:  Performance Analysis of Photovoltaic Integrated Shading Devices (PVSDs) and Semi Transparent Photovoltaic (STPV) Retrofitted to a Prototype Office Building in a Hot Desert Climate

REVIEWER (3) COMMENTS

ACTION TAKEN

Reviewer (3) Comments

1

Review the list of authors: I think there is a lack of "," between the first two authors and it seems that there are surnames in lower case

·         The comma has been added between two authors.

2

Line 218: is the reference to table 4 correct? Authors should check the numbering of all tables and figures and their references in the text.

·         - Thank you for your comment, yes the table 4 has been included:

Table 4: the performance indicators of visual comfort used in this study

Analysis

Criteria

Performance Indicator

Quantitatife

+

qualitatife

UDI

100 lux < Dark area ( need artificial light)

100 lux – 2000 lux (comfortable), at least 50% of the time

>2000 lux Too bright with thermal discomfort

DA

Set up 300lx

WPI

WPI Recommended 300-500 lux

DGP

0.35 < Imperceptible glare

0.35 - 0.40 perceptible glare

0.4 – 0.45 Disturbing glare

>0.45 Intolerable glare

·          

·         - All tables numbering has been revised

4

Line 239: the authors could explain the higher levels of energy production on horizontal surfaces in summer by referring to the position of the sun.

·         The statement of sun path position has been added to concrete the result:

·         “the 0⁰-tilt PV produce more electricity overall as it had a southern orientation and due to the sun path position, therefore, received more solar radiance”.

5

Line 292: a reference to a table 5 that does not exist in the document is made. Does it refer to table 4?

·         The numbering of tables has been rectified

6

Table 4: the pictures included in this table are too small and cannot be clearly understood.

Also, the explanatory texts in the first columns of the table cannot be seen properly.

All 3D illuminance contour map has been modified to be clear (table 5) after update the numbering.

The texts in table 5 have been modified.

·          

7

Figure 1: For a better understanding of the working scenario, figure 1 should appear before table 1.

·         The correction has been rectified, the climatic condition first then the explanation of PVSD configuration.

8

Figure 2: the source of the data must be indicated; the vertical axis lists magnitudes that are not represented in the graph.

·         The figure2 of climatic condition has been improved.

9

Figure 4: the authors say that they represent energy but the unit indicated on the vertical axis (watts) is not a unit of energy (but of power). The authors should review the energy units also in the paragraph before this figure.

·         - The unit has been rectified (watt per hour) which is represent the energy output.

10

The comparison of the influence of tilt angles would be clearer if it referred to the same day

·         - Yes, the comparison of PV module with different tilted angles (horizontal, 30o, 60o and vertical) has been conducted at the same day and within four design days (solstice summer, equinox autumn, solstice winter and equinox spring).

11

Figure 5: Authors should specify whether the data represented are simulated or real.

The authors could explain the shape of the graph on 21 June.

· - The data used in this graph is through simulation. (please check the pdf file)

·         - The value of indoor illuminance (lux) reduced in the evening period of 21 of June because of reduction of external illuminace, where sky condition changed from clear sky to overcast sky. Here the indoor illuminance simulation, the day before and after (20 , 21 and 22 June) show the pattern:

·        

12

Figure 8: the results shown in this figure and the conclusions drawn from them need to be further explained

The explanation of this part has been added

-As seen in Figure 8, the ROB had the lowest mean DGP; the perceptible and disturbing glare mainly appear at winter season between 10am till 3pm, the DGP value is between 27 to 40; over the four assessment days

- Overall, the configurations considerably enhanced visual comfort with a minimum 2° reduction in mean DGP compared to the ROB. Hence we can claim that PVSDs application are suitable for winter season while STPV with 20% of transparency is totally eliminate the glare issue but do not provide sufficient indoor illuminance.

13

Figure 9: The efficiency percentages shown in the figure do not correspond to those quoted in the text (lines 311-312).

·         - The efficiency percentages has been corrected in the paragraph:

"Figure 9 presents the energy saving potential of the PVSDs at varying efficiencies (ȵ10%, ȵ15% and ȵ20%) and the STPV module at ȵ3.1%".

Round 2

Reviewer 4 Report

Authors have improve the quality of the paper. However, I recommend the authors to make two corrections that are still to be made from the first review:
Figure 4: It would be useful to represent all cases for the same day. In this way, it would be easier to compare the influence of different tilt angles on the received irradiance.
Figure 5: Authors should include a brief explanation justifying the shape of the curve on June 21st.

Author Response

REVIEWER COMMENTS

ACTION TAKEN

Reviewer

Figure 4: It would be useful to represent all cases for the same day. In this way, it would be easier to compare the influence of different tilt angles on the received irradiance.

The main purpose of the figure4 is to present a validation of measured and simulated data; however it is an experiment results data, so we cannot compare all data at the same day. Thus, In order to compare the influence of different tilt angles on the received irradiance. Figure5 clarified the various scenarios in four design days.

Figure 5: Authors should include a brief explanation justifying the shape of the curve on June 21st.

The brief od explanation has been added to the point: 3.1 Analysis and Validation of energy Outputs at Different Tilt Angles

It is remarkable that the value of indoor illuminance (lux) reduced in the evening period of 21 of June because of reduction of external illuminace, where the sky condition changed from clear sky to an overcast sky.
